# Tropospheric Ozone sensing with a differential absorption lidar based on single $CO_2$ Raman cell

Guangqiang Fan,[1] Yibin Fu,[1] Juntao Huo,[3] Yan Xiang[4], Tianshu Zhang,[1,2] Wenqing Liu[1] and Zhi Ning[5]

[1]Key Laboratory of Environmental Optics and Technology, Anhui Institute of Optics and Fine Mechanics, Chinese Academy of Sciences, Hefei, 230031, China

[2] Institute of Environment, Hefei Comprehensive National Science Center, Hefei 230088, China

[3]Shanghai Environmental Monitoring Center, Shanghai, 200235, China

[4]Information Materials and Intelligent Sensing Laboratory of Anhui Province, Institutes of Physical Science and Information Technology, Anhui University, Hefei 230601, China

[5]Division of Environment and Sustainability, The Hong Kong University of Science and Technology, Hong Kong SAR, China

*Correspondence to*: Yan Xiang(yxiang@ahu.edu.cn) and Tianshu Zhang (tszhang@aiofm.ac.cn)

**Abstract.** This study presents the development and performance evaluation of an ozone differential absorption lidar system. The system could effectively obtain vertical profiles of lower tropospheric ozone in an altitude range of 0.3 to 4 km with high spatiotemporal resolutions. The system emits three laser beams at wavelengths of 276 nm, 287 nm, 299 nm by using the stimulated Raman effect of carbon dioxide ($CO_2$). A 250 mm telescope and a grating spectrometer are used to collect and separate the backscattering signals at the three wavelengths. Considering the influences of aerosol interference and statistical error, a wavelength pair of 276 nm-287 nm is used for the altitude below 600 m and a wavelength pair of 287 nm-299 nm is used for the altitude above 600 m to invert ozone concentration. We also evaluated the errors caused by the uncertainty of the wavelength index. The developed ozone lidar was deployed in a field campaign that was conducted to measure the vertical profiles of ozone using a tethered balloon platform. The lidar observations agree very well with those of the tethered balloon platform.

## 1 Introduction

Tropospheric ozone is an important greenhouse gas and plays an important role in the Earth's radiation budget with an estimated direct radiative forcing of 0.4 W m$^{-2}$ during the industrial era(X. Chen et al.,2020). At the surface, ozone is an important air pollutant that impacts the oxidative capacity of the atmosphere(T. Wang et al.,2017). It is highly reactive with oxidative potential to damage biological tissues and adversely impacts human health, vegetation, crop yield and crop quality. As a result of ozone's high reactivity, the lifetime of ozone in the lower troposphere is short with significant differences in spatial and temporal distributions. For a specific region, tropospheric ozone mainly originates from the photochemical production of local anthropogenic and biogenic emissions(B. Koo et al.,2012; X. Chi et al.,2018), regionally advection transport(E. Schuepbach et al.,1999; X. Wang et al.,2021) and stratosphere-troposphere exchange(G. Clain et al.,2010; M. A.

Olsen et al.,2004; M. Wang et al., 2023). With such dynamic sources, it is essential to monitor both vertical and temporal distributions of tropospheric ozone for making effective control strategies of ozone pollution.

At the surface, in-situ ultraviolet analyzers can measure ozone concentrations with high temporal resolutions and high accuracy of within 5%(J. T. Sullivan et al.,2014). A national network of surface ozone monitoring has been gradually established covering nearly all the cities in China over the past few decades. The measurements of these surface stations are typically in 8-hour average or hourly average values which are effective for analyzing surface ozone trends. However, it is essential to analyze vertical variations of lower tropospheric ozone when dramatic changes of surface ozone occurred. There are several useful methods, including tethered balloon(X. B. Li et al.,2018), sounding balloon(M. A. Lokoshchenko et al.,2009) and aircraft(A. O. Langford et al.,2019), that have been successfully used to obtain vertical profiles of lower tropospheric ozone. However, the measurements made by these methods have limited spatial and temporal variations and cannot fully characterize the distribution and evolution patterns of ozone in the lower troposphere. Ozone profiles from the tropospheric Emission Spectrometer and the Ozone Mapping Instrument have been reported(G. B. Osterman et al.,2008; H. M. Worden et al.,2007; Y. Qian et al., 2021), while the vertical resolution for tropospheric ozone is strictly limited.

The continuous vertical and temporal distributions of ozone in the troposphere can be detected by differential absorption lidars with much higher frequency and accuracy. Dating back to the 1970s, this technique was first used to monitor water vapor. The technique was then successfully modified and utilized for accurate ozone detection. For ozone detection, the differential absorption lidars can be divided into two groups according to the types of laser technology: tunable laser technology and fixed frequency conversion technology. The advantage of tunable laser technology for ozone detection relies on its optimal detection wavelengths, which contributes to optimal detection sensitivity improvements including small aerosol and other gas interference. The second harmonic of Nd:YAG laser pumps the dye laser and produces a series of waves in the range of 272 nm~310 nm via the double frequency crystal as the light source of ozone detection. Most ozone lidars use two separate dye lasers to generate both on and off wavelength pairs at the same time(S. Kuang et al.,2013). The differential absorption lidar based on this technique usually has a complex system and requires a wavelength stabilization feedback device to monitor and control the laser wavelength in real time. In addition, dye lasers need to be replaced frequently due to a limited life time of the dye, some of which contain carcinogens and are harmful for the operators' health. Given these drawbacks, for the purpose of optimizing ozone detection, researchers have developed a pulsed optical parametric oscillator with intracavity sum frequency mixing, generating lasers in the wavelength range of 281-293 nm(A. Fix et al.,2002).

Moreover, excimer lasers or Nd:YAG lasers are also used in some studies as the pumping lasers. $H_2$ and $D_2$ Raman gases are pumped to produce stokes lights for ozone detection(I. D. Hwang et al.,1993; G. Fan et al., 2024). National Oceanic and Atmospheric Administration (NOAA) deployed a scanning four-wavelength ultraviolet differential absorption lidar(J. L. Machol et al.,2009). The lidar system measures tropospheric ozone and aerosols by utilizing the Raman shift wavelengths generated from $D_2$ and $H_2$ gases. However, there are two primary challenges associated with employing the $D_2$ and $H_2$ dual Raman cells: (1) Shared Laser Resource: The $D_2$ and $H_2$ Raman cells are both pumped by the same frequency-quadrupled

Nd:YAG laser. This shared resources places increased demands on the pump laser's performance and stability. The lase must provide sufficient energy to effectively pump both Raman cells. (2) Receiver Field of View and Laser Divergence Overlap: The second challenge arises from the varying overlaps between the receiver's field of view and the divergences of the laser beams for the $D_2$ and $H_2$ Raman cells. These differences can result in a larger blind area during ozone detection. The blind area refers to the region where the lidar system is unable to accurately measure ozone concentrations due to the geometric constraints of the laser beam and the receiver's field of view. This can lead to incomplete or inaccurate data regarding the ozone levels in the troposphere. In contrast, an ozone lidar system utilizing a $CO_2$ single Raman cell has the potential to address these issues. The single Raman cell design simplifies the system by eliminating the need to manage two separate Raman cells, thereby reducing complexity and the need for Nd:YAG. Furthermore, the single Raman cell system may offer a more consistent overlap between the receiver's field of view and the laser beam, which can help to minimize the blind area and enhance the accuracy of ozone detection.

However, until now, only few studies developed the ozone lidar using a single $CO_2$ Raman cell to detect ozone in both the planetary boundary layer and the free troposphere simultaneously. Many uncertainties including aerosol interference induced errors, and the system errors caused by wavelength index uncertainty are worth for researchers to conduct a more thorough investigation.

In this paper, we present an ozone differential absorption lidar system based on the single $CO_2$ Raman cell and the grating spectrometer. The wavelength selection and theoretical analysis of aerosol interference errors are discussed in Section 2. The design and architecture of the ozone lidar are introduced in detail in sections 3. Analysis of statistical errors and the system errors caused by Angstrom wavelength index uncertainty are discussed in sections 4 and 5, respectively. Finally, the section 6 provides a typical field validation for the ozone lidar developed in this study by using ozone vertical observations of a tethered balloon platform.

## 2 Theoretical analysis

According to the dual-wavelength differential absorption algorithm, the ozone concentration $N(z)$ can be expressed as follows(S. I. Dolgii et al.,2017):

$$N(z) = \frac{1}{2\Delta\delta} \frac{d}{dz}[-\ln(\frac{P(\lambda_{on},z)}{P(\lambda_{off},z)})] + B - E_a - E_m - E_{gas} \tag{1}$$

$$B = \frac{1}{2\Delta\delta} \frac{d}{dz}[\ln(\frac{\beta(\lambda_{on},z)}{\beta(\lambda_{off},z)})] \tag{2}$$

$$E_a = \frac{1}{\Delta\delta}[\alpha_a(\lambda_{on},z) - \alpha_a(\lambda_{off},z)] \tag{3}$$

$$E_m = \frac{1}{\Delta\delta}[\alpha_m(\lambda_{on},z) - \alpha_a(\lambda_{off},z)] \tag{4}$$

$$E_{gas} = \frac{\Delta\delta_{gas} N'_{gas}}{\Delta\delta} \tag{5}$$

Where $P(\lambda_i, z)$ is the atmospheric backscatter echo signal at wavelength $\lambda_i$ and range $z$; $i$ is on or off; $\Delta\delta = \delta_{\lambda_{on}} - \delta_{\lambda_{off}}$ is the differential absorption cross section of ozone; Where $B$, $E_a$ and $E_m$ are the systematic errors from atmospheric backscattering, aerosol extinction and molecular extinction; $E_{gas}$ is the systematic error introduced by the absorption effect of other trace gasses; $\beta(\lambda_i, z)$ is total atmospheric volume backscatter coefficient at wavelength $\lambda_i$ and range $z$; $\alpha(\lambda_i, z)$ is total atmospheric optical extinction coefficient neglecting ozone absorption at wavelength $\lambda_i$ and range $z$; $\Delta\delta_{gas}$ is the differential absorption cross section of other trace gases; $N'_{gas}$ is the concentration of other trace gases; $\alpha_a$, $\alpha_m$ are respectively the extinction coefficients of atmospheric particulate matter and molecular, respectively.

The distribution of molecules in the atmosphere is stable, exhibiting less variable. Therefore, the atmospheric molecular extinction coefficient is directly used to correct $E_m$. Generally, $B$ and $E_a$ cannot be neglected in the measurements of boundary layer ozone when using the differential absorption method due to that the atmospheric backscattering and aerosol extinction coefficients exhibit strong wavelength dependence. Given that $\Delta\lambda = \lambda_{off} - \lambda_{on}$ is small, the aerosol extinction correction $E_a$ and the backscatter correction $B$ can be estimated using the following equations(M. Nakazato et al.,2007):

$$E_a \approx -\frac{\Delta\lambda}{\Delta\delta\lambda_{off}} k\alpha_a(\lambda_{off}, z) \tag{6}$$

$$B \approx \frac{(4-\mu)}{2dz\Delta\delta} \cdot \frac{\Delta\lambda}{\lambda_{off}} \cdot [\frac{S_{off}(r)}{1+S_{off}(r)} - \frac{S_{off}(r+dz)}{1+S_{off}(r+dz)}] \tag{7}$$

$$SF = \frac{1}{\frac{\Delta\delta}{\Delta\lambda}\lambda_{off}} \tag{8}$$

Where $k$ and $\mu$ are the power-law exponents for backscattering and extinction, $S_{off}(r)$ is the aerosol backscatter ratio, and $SF$ is referred to as the spectrum factor or the differential aerosol backscatter sensitivity. $E_a$, $B$ are proportional to $SF$. $E_a$ is proportional to $k$, and $B$ is proportional to $(4-\mu)$. As reported in previous studies, the angstrom wavelength index was generally in the range of 0.6 to 1.4 and exhibited strong spatial and temporal variations Therefore, it is assumed that the values of $k$ and $\mu$ vary in this range. However, the values of $k$ and $\mu$ were assumed to be 1 when calculating aerosol correction terms using measured data. Due to the changes of $k$ and $\mu$, $E_a$ was in error of 40%; $B$ error is within 13%. The aerosol interference is inevitable if the values of $k$ and $\mu$ are uncertain, which makes it crucial for the choice of $SF$. Theoretically, the smaller the $SF$ is, the smaller the influence of aerosol interference on ozone retrieval results.

**Table 1 *SF* of the differential absorption wavelength pairs**

| wavelength pairs (nm) | $\Delta\delta$ (e$^{-20}$ cm$^2$) | $\Delta\lambda$ (nm) | $SF$ (e$^{-16}$ cm$^2$) |
|---|---|---|---|
| 289/316.4 | 152.61 | 27.4 | 5.67 |
| 299.1/316.4 | 39.79 | 17.3 | 13.74 |
| 289/299.1 | 112.82 | 10.1 | 3.01 |
| 266/289 | 773.4 | 23 | 1.03 |
| 276.2/287.2 | 335.43 | 11 | 1.14 |
| 287.2/299.1 | 152.21 | 11.9 | 2.61 |

The Nd:YAG quad-frequency laser, when used to pump a single $D_2$ Raman tube, generates both first-order and second-order Stokes light. These correspond to the differential absorption wavelengths of 289 nm and 316 nm, respectively. The pumped $D_2$ Raman tube and $H_2$ Raman tube produce first-order stokes light, corresponding to the differential absorption wavelengths of 289 nm/299 nm, respectively. 289 nm/316 nm, 289 nm/299 nm, and 266 nm/289 nm are common differential absorption wavelength pairs for ozone retrieval. Nd: YAG quad-frequency laser pumped single $CO_2$ Raman tube generates first-order stokes light, second-order stokes light and third-order stokes at corresponding differential absorption wavelengths of 276.2 nm, 287.2 nm and 299.1 nm. Table 1 lists the SF of the differential absorption wavelength pairs. The $SF$ of the differential absorption wavelength pair of 276.2 nm and 287.2 nm is nearly half of that of the 287.2 nm and 299.1 nm pair, indicating that $E_a$ , $B$ of the wavelength pair of 276.2 nm and 287.2 nm is nearly half of that of the wavelength pair of 287.2 nm and 299.1 nm. Due to the strong absorption of ozone at 276nm and the strong atmospheric backscatter at this wavelength, the detection height of the 276nm wavelength signal is limited. As shown in Fig. 1, the signal-to-background ratio of the 276nm signal is greater than 100 below 600m, which meets the detection requirements with sufficient signal-to-noise ratio. Above 800m, it quickly drops below 100. To accommodate different aerosol types and weather influences, and considering that aerosols are mainly distributed below a height of 600m, a height of 600m was adopted as the stitching height for the differential wavelength pair. Above 600m, we adopted the wavelength pairs of 287.2 nm and 299.1 nm for ozone detection. The $SF$ of the wavelength pair of 287.2 nm/299.1 nm is slightly smaller than the wavelength pair of 289 nm/299.1 nm, which is the most widely used wavelength pair in gas Raman tube technology, indicating that $E_a$ , $B$ of 287.2 nm and 299.1 nm is smaller. The $SF$ of the wavelength pair of  287.2 nm/299.1 nm is half of the wavelength pair of 289 nm and 316.4 nm, so the aerosol interference term is half that of the wavelength pair of 289 nm and 316.4 nm.

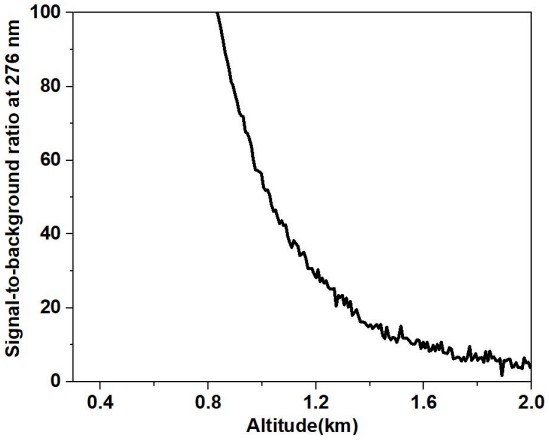

 **Fig. 1. The signal-to-background ratio at 276 nm wavelength**

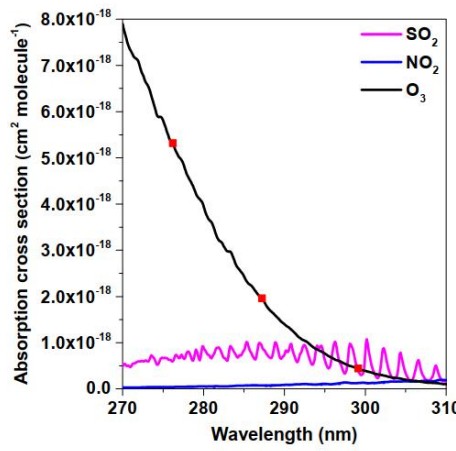

**Fig. 2. The absorption cross sections of $O_3$, $NO_2$, and $SO_2$ at 276.2nm, 287.2nm and 299.1nm**

**Table 2 The interference of $NO_2$ and $SO_2$**

| Wavelength pairs (nm) | $SO_2(e^{-20}cm^2)$ | $NO_2(e^{-20}cm^2)$ | $O_3(e^{-20} cm^2)$ | $NO_2$ interference | $SO_2$ interference |
|---|---|---|---|---|---|
| 276.2/287.2 | 30 | 3.3 | 335.43 | 0.98% of the $NO_2$ concentration | 8.9% of the $SO_2$ concentration |
| 287.2/299.1 | 52.8 | 5.4 | 152.21 | 3.5% of the $NO_2$ concentration | 34.7% of the $SO_2$ concentration |

According to formula (5), the influence of $NO_2$ and $SO_2$ on the $O_3$ retrieval can be determined. Figure 2 shows the absorption cross sections of $O_3$, $NO_2$, and $SO_2$ at 276.2nm, 287.2nm, and 299.1nm. The table 1 below analyzes the extent of interference

from NO$_2$ and SO$_2$ gases. The interference from NO$_2$ at the 276.2 nm/287.2 nm wavelength pair and the 287.2 nm/299.1 nm wavelength pair is 0.98% and 3.5% of the NO$_2$ concentration, respectively, which can be neglected. The impact of SO$_2$ on ozone is more significant, with impacts of 8.9% and 34.7% of the SO$_2$ concentration at the two wavelength pairs.The typical environmental concentration of SO$_2$ is a few (ug/m³). If assessed at 10 ug/m³, its impact would be approximately 0.89 ug/m³ and 3.5 ug/m3, which is relatively small compared to other sources of error and is therefore usually not considered.

## 3 Ozone lidar system architecture

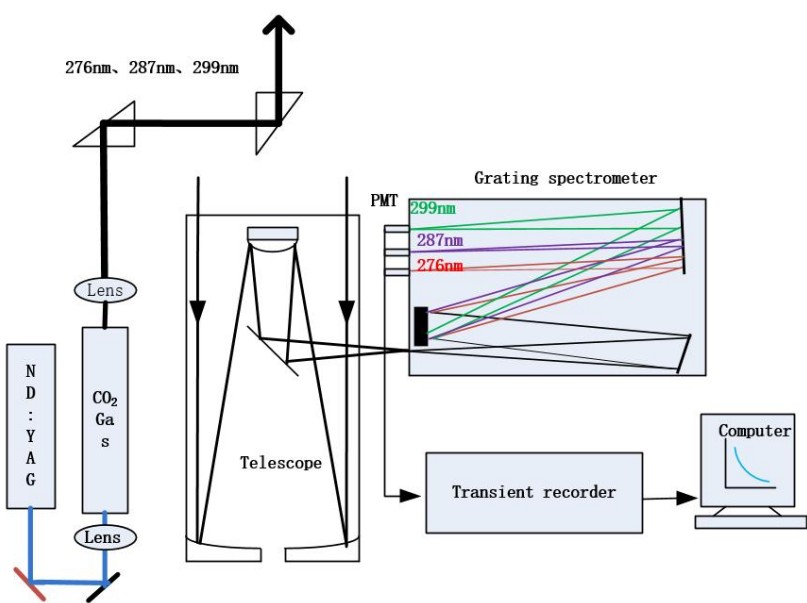

Fig. 3. Schematic diagram of the ozone lidar system based on the single CO$_2$ Raman cell.

We designed a differential absorption lidar based on a single CO$_2$ Raman cell for measuring boundary layer and free tropospheric ozone. Compared with the D$_2$ and H$_2$ Raman tube, this system has a smaller SF to reduce the aerosol interference , which makes it particularly suitable for the detection of lower tropospheric ozone. Figure 3 shows the schematic diagram of the ozone lidar system developed in this study. The key parameters of the ozone lidar system listed in Table 2. The ozone lidar is mainly composed of three parts: laser transmitting unit, optical receiving and subsequent optical unit, and data acquisition unit. The whole system is based on the same optical plate with a compact and stable mechanical structure.

Table 2. The key parameters of the differential absorption lidar system

| Cell name | Parameters |
|---|---|
| **Transmitter** | |

| | |
|---|---|
| Pumped Laser | Nd: YAG (266 nm) |
| Pulse repetition rate | 10 Hz |
| Pulse Energy | 90 mJ |
| Raman Shifted wavelength | 276 nm, 287 nm, 299 nm |
| Output energy | 8.4 mJ, 7.7 mJ,4.2 mJ |
| Divergence angle | 0.3 mrad |
| **Receiver** | |
| Telescope | Cassegrain |
| Telescope diameter | 250 mm |
| Detector | Photomultiplier tube |
| Data acquisition | Analog digitizer |
| **System parameters** | |
| Detection range | 0.2 km~4 km |
| Temporal resolution | 15 min |
| Spatial resolution | 75 m |
| Statistical error | <15%(below 3 km) |

### 3.1 Laser transmitting unit

A flashmap-pumped Nd: YAG laser (Quantel, Q-Smart 850), which provides 90 mJ output at the wavelength of 266 nm and the pulse repetition rate of 10 Hz, is used as the pump source for the $CO_2$ Raman cell. Considering the volume of the final equipment and the $CO_2$ stimulated Raman optimization experiment, the Raman cell with a length of 1 m is adopted. The Raman cell is filled with 16 bar pure $CO_2$ with 99.999% purity. It has high strength and good tightness. The 266 nm laser is focused near the center of the Raman cell with a 15 mm inner diameter using a 500 mm focal-length lens. Raman cell incident lens and achromatic lens group constitute a triple beam expansion system, and the laser divergence angle is 0.3 mard. The energy output of the laser at wavelengths of 276 nm, 287 nm, and 299 nm are 8.4 mJ, 7.7 mJ, and 4.2 mJ, respectively. The purpose of adopting achromatic lens is to minimize the difference of the laser divergence angle at the wavelengths of 276 nm, 287 nm and 299 nm, to reduce the influence of geometric factors in lidar transition zone on ozone retrieval, and to increase the lower detection height of the ozone lidar. The arrangement of coaxial transmission and reception was also adopted to further reduce the maximum height of the lidar transition zone.

### 3.2 Optical receiving and subsequent optical unit

This system deployed a Cassegrain telescope with a diameter of 250 mm and a focal length of 2500 m. The primary and the secondary mirrors of the Cassegrain telescope are hyperboloid mirrors. The telescope is mounted on a rigid optical bench

together with the laser transmitting unit. An ultraviolet multiwavelength grating spectrometer is used to separate the echo signals at the wavelengths of 276 nm, 287 nm and 299 nm. The grating spectrometer includes an aperture, a high reflection collimator, a high resolution holographic grating, three sets of high reflectivity plano-concave reflectors, and three sets of photomultiplier tubes. These components are mounted on an optical plate and sealed by a closed black box to avoid the light interference. The 2 mm aperture is mounted on the focal plane of the receiving telescope and the received field view angle of the ozone lidar system is about 0.5 mrad. The echo signals at the wavelengths of 276nm, 287nm and 299nm are converged by the receiving telescope to form a divergent beam with a numerical aperture of 10. When the signals are transmitted to the grating spectrometer, the lights are collimated by the plano-concave mirror. Through the reflection of the plano-concave mirror, the parallel light arrives at the diffraction grating. The high resolution planar holographic grating is the core part of the grating spectrometer. Echo signals at the wavelengths of 276 nm, 287 nm and 299 nm can be separated into different angle positions due to their different diffraction angles. The three sets of high-reflection flat concave mirrors are constructed using JGS1 quartz material, which is chosen for its superior optical properties and resistance to laser damage, ensuring high reflectivity and durability in the system. The plating of high-reflection dielectric film increases the reflectivity to more than 98% for the optical signals in the ultraviolet band. By adjusting the angles of the three sets of the high-reflection flat concave mirrors, the echo signals could precisely converged in the three sets of photomultiplier tubes. R7400 photomultiplier tubes produced by Hamamatsu is applied, with an effective receiving aperture of 8 mm, short response time and high quantum efficiency in the ultraviolet band of 200 nm~300 nm.

### 3.3 Data acquisition unit

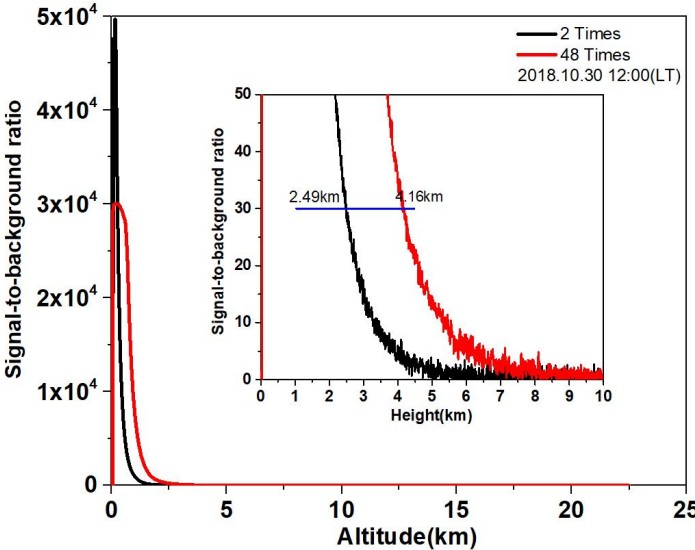

Fig. 4. The echo signal at 287nm wavelength and the signal-to-background ratio.

A 20 M, 12-bit A/D data acquisition system, is selected to record single shot raw data, providing a vertical spatial resolution of 7.5 m that is good enough for ozone measurements with vertical resolution ranging from 75 to 200 m. The maximum number of samples is set as 3000 to monitor the sky background noise. Therefore, various background baseline distortions due to the presence of electromagnetic interference or SIN effects in the tail of the lidar signals can be monitored. The echo signals are averaged for as many as 4000 shots (400 s acquisition time) by the software. In order to meet the long-term

monitoring requirements and increase the service life of the laser, we stop the flash lamp for 300 s after each echo signal averaging. Thus, the time resolution of the system is 700 s. In order to reduce the influence of the A/D electronic noise on the atmospheric echo signal, amplifiers of 2 times and 48 times are adopted for the same echo signal, respectively for the low altitude signal and the high altitude signal. The signal obtained from 15 to 16 km is selected as the background signal, the standard deviation is calculated as the electronic noise. Taking the echo signal of 287 nm as an example, the influence of

different amplifier magnification times on the effective detection of the signal is illustrated in Fig. 4, with the echo signal at 287 nm and the signal-to-background ratio defined as the ratio of the echo signal to the standard deviation of the background signals. Below 500 m, the signal collected by the acquisition card is saturated with the 48 times amplification of the 280 nm echo signal. The detection heights of the 2 times amplified and 48 times amplified signal are 2.49 km and 4.19 km, respectively, when the signal-to-background ratio (SBR) is 30. It can thus be seen that a large magnification can effectively

increase the dynamic range of the echo signal as well as improve the detection range of the echo signal. Within the range of 0.5~1 km, the echo signals of the 2 times and 48 times magnification are fused using the least square method.

## 4 Inversion errors analysis

### 4.1 Analysis of statistical error

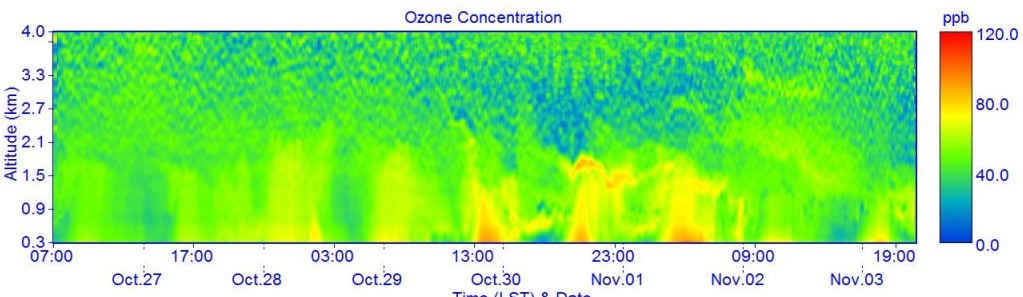

**Fig. 5. Time series of ozone vertical profiles obtained from the ozone lidar between October 26 to November 3, 2018.**

    The ozone lidar was initially located at the Anhui Institute of Optics and Fine Mechanics in Hefei, China. The experiment was first conducted from October 26 to November 3, 2018, as shown in Fig. 5. Below 600 m, the signals at 276 nm and 287 nm were used to retrieve ozone concentration profile; Above 600 m, the signals at 287 nm and 299 nm were used. This image was created by analyzing the measurement results from the emitting 4000 laser pulses to construct a complete profile

of the atmosphere from 0.3 to 4 km with a vertical resolution of 100 m. During the observation period, ozone concentrations below 2 km exhibits a distinct diurnal distribution pattern and experienced the processes gradual accumulation, aggravation, and dissipation. High ozone mixing ratios of exceeding 60 ppb occurred in most of the afternoon periods from October 30 to November 1. The statistical error of ozone lidar data is inversely proportional to the absorption cross-section difference, the difference distance, the unknown gas concentration, and the SNR of the ozone data. The statistical error of the ozone lidar is

related not only to the hardware of the device but can also be considered constant in the short term, aside from its dependence on atmospheric conditions and solar irradiance. Generally, due to the influence of solar irradiance, the signal-to-noise ratio (SNR) of daytime signals is typically lower than that of nighttime signals. During the observation period from October 26 to November 3, 2018, the SNR of the 299.1nm signal remained essentially stable as shown in Fig. 6. Therefore, the statistical error of ozone at 11:00 to 12:00 on October 26, 2018, was used to analyze the performance of the ozone lidar.

It is important to note that an ozone lidar is an in situ measurement device that is closely related to atmospheric conditions, and its SNR can drop sharply during extreme weather conditions such as rain or fog, leading to a significant increase in statistical error. Six profiles measured by the ozone lidar from local standard time (LST) 11:01 to 12:03 (LST) on October 26, 2018 were selected for statistical analysis. As shown in Fig. 7, the statistical error in the height range of 200 m to 600 m gradually increased from 2.35% to 6.9% and it was accompanied by the decrease in the ratio of signal to noise for the echo

signals of 276 nm. The statistical error of ozone from 0.6 to 2.7 km was basically within 3%. From 2.7 km to 3.9 km, the statistical error gradually increased to about 18%, which is due to the gradual deterioration of the signal-to-background ratio with the increase of height.

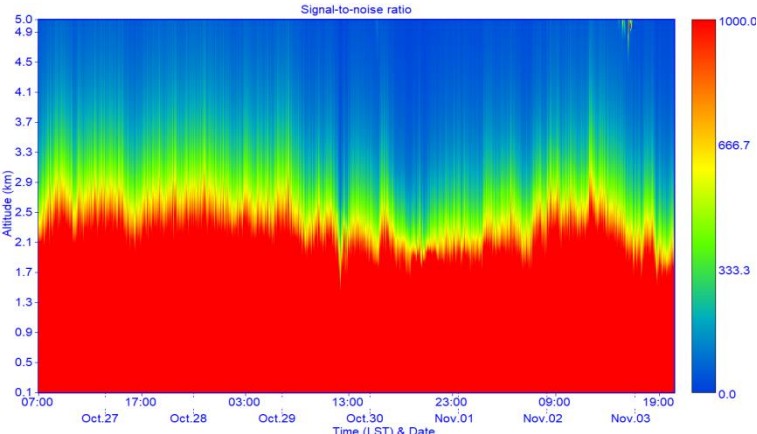

**Fig. 6. Signal-to-noise ratio of the echo signal at 299.1nm wavelength**

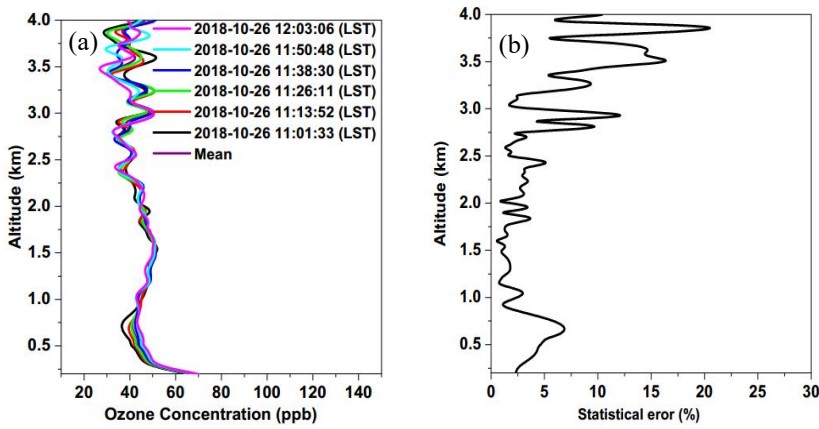

**Fig. 7. Ozone vertical profile measured by the ozone lidar and its statistical error.**

**4.2 Analysis of system errors caused by wavelength index uncertainty**

The signals at 299 nm is used to retrieve the aerosol backscattering coefficient and extinction coefficient due to that the ozone absorption at this wavelength was negligible. In addition, the signals at 299 nm is also used to obtain real-time correction terms for aerosol extinction and backscattering. Figure 8 shows the time series of the vertical profiles of aerosol extinction coefficient at 299 nm obtained by the ozone lidar between October 26 to November 3, 2018 at a time resolution of 12 min. During the observation period, the boundary layer height had an obvious diurnal variation pattern before October 30.

The boundary layer height was about 2 km and the aerosol extinction coefficient was lower than 0.3 km$^{-1}$. The boundary layer height decreased from October 31 to November 3, during which the maximum boundary layer height was about 1.4 km. Meanwhile, the concentration of particulate matter in the boundary layer increased significantly and the maximum aerosol extinction coefficient was 1.2 km$^{-1}$. From October 31 to Novermber 1, the downward transport of aerosols occurred within the height range of 1.4 km to 2 km. During the observation period, the aerosols in the boundary layer and free troposphere

had distinct distribution patterns with relatively higher concentration levels about from 0.3 km$^{-1}$ to 1 km$^{-1}$ in the boundary layer. The vertical observations of the aerosol extinction coefficient could also effectively capture the transport of aerosol in the lower free troposphere, as shown in Fig. 8. In addition, the aerosol observations could be used to study the influence of the spatial variations and aerosol concentration levels on the uncertainty of ozone retrieval.

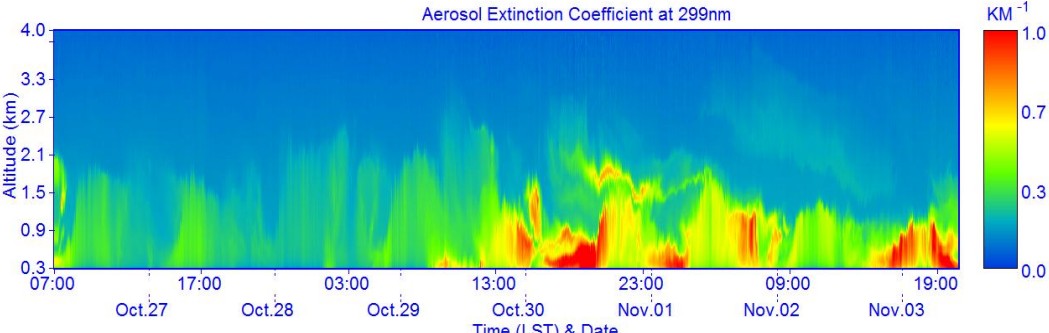

**Fig. 8. Time series plot of the aerosol extinction coefficient at 299nm from the ozone lidar between October 26 to November 3, 2018 at a 12 min temporal resolution.**

Aerosol correction term ( $E_a + B$ ) was shown in Fig. 9. Values of the aerosol correction was small when the aerosol extinction coefficient was lower than 0.5 km$^{-1}$ before October 30. However, when the aerosol concentration sharply increased and strongly varied (such as from October 31 to November 3), particularly on the boundary layer top and during the aerosol transport process, the aerosol correction term also increased suddenly, often exceeding 15 ppb, which cannot be ignored.

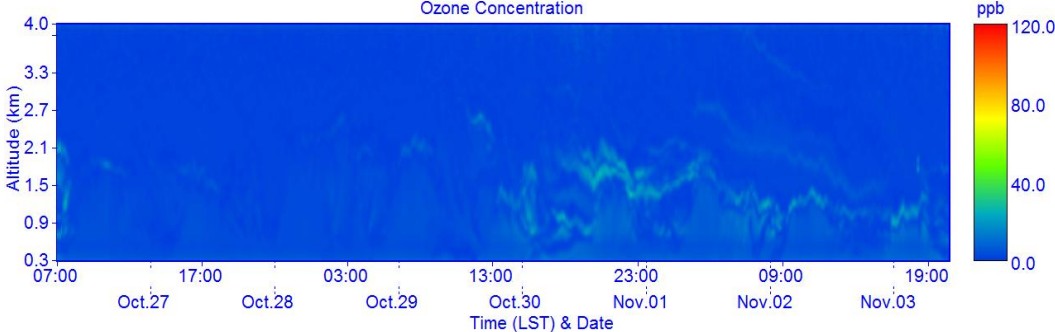

**Fig. 9. Time series of the vertical profiles of the aerosol correction term ( $E_a + B$ ) between October 26 to November 3, 2018.**

Figure 10a shows the aerosol correction term ( $E_a + B$ ) obtained at 300 m, 500 m, 1500 m, 2000m, respectively, from October 26 to November 3, 2018. The 300 m height is basically within the boundary layer and the aerosol correction term fluctuated around 10 ppb. Before October 30, the aerosol correction terms were below 5 ppb at altitudes of 500 m, 1000 m, 1500 m, and 2000 m. From October 30 to November 3, when the aerosol concentration in the boundary layer was high and the aerosol transportation was found outside the boundary layer from1.4 km to 1.8 km, the aerosol correction terms changed dramatically which was consistent with the boundary layer characteristics. The maximum value of the aerosol correction term reached about 20 ppb. The vertical distribution characteristics of aerosol correction terms were analyzed at typical sampling periods. As shown in Fig. 10b, at LST 18:00 on October 27, and LST 17:58 on October 29, 2018, the aerosol concentrations were relatively low, and the aerosol correction terms decreased with the increase of height between 0.3 and 3.5 km. The aerosol correction terms were about 10 ppb at 300 m. Above 500 m, it rapidly dropped to below 4ppb and

became smaller with the increase of height, which had little influence on the retrieval of ozone. At LST 17:58 on October 31, the vertical profile of aerosol correction term also changed dramatically between 1.5 and 2.2 km, resulting in a bimodal distribution pattern. In the boundary layer where the aerosol concentration was high, the aerosol correction term also exhibited a bimodal distribution pattern with dramatic changes from the lowest level of 4 ppb to 14 ppb. The analysis indicates that the aerosol correction term exhibits rapid fluctuations during the transport of aerosols, particularly when the concentration of boundary layer aerosols is elevated. Therefore, it is necessary to correct the ozone retrieval results in real time using the aerosol correction term.

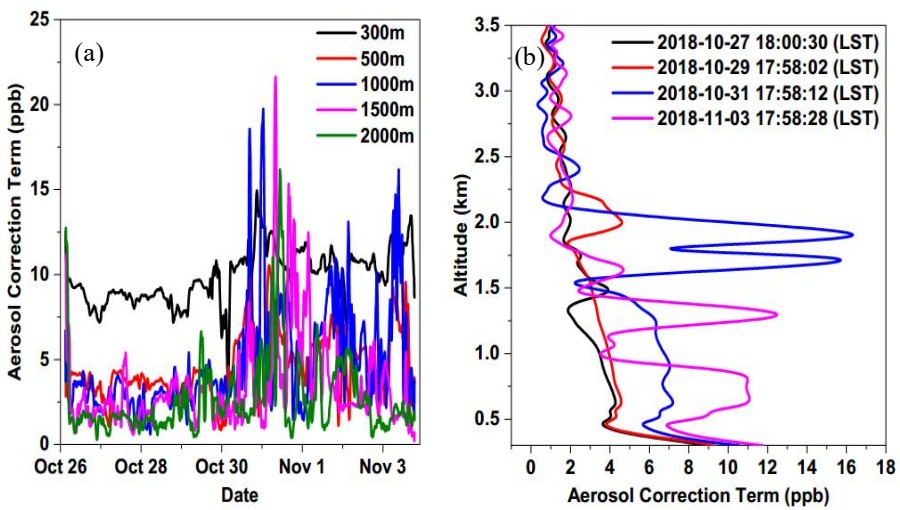

**Fig. 10. Aerosol correction terms at different height and time between October 26 to November 3, 2018.**

Figure 11 shows the errors of aerosol correction terms caused by changes in $k$ and $\mu$ from 1 to 0.6. From formula (2) it can be seen that when the variation of $k$ and $\mu$ of both were 0.4, the resulting aerosol correction term errors were basically the same, and the maximum aerosol correction term errors caused by dramatic changes in aerosol was about 5 ppb. Figure 12 shows the aerosol correction term errors of $k$ and $\mu$ from 1 to 1.4 at different heights and times. Before October 30, the errors of aerosol correction term in the range of 300 m~3.5 km were all less than 2 ppb. At 17:58 (LST) on October 31, 2018, the maximum error of aerosol correction term was 5 ppb during aerosol transport between 1.5 and 2.2 km; the error showed a single-peak distribution pattern. The aerosol correction error is acceptable for ozone monitoring and meets the detection requirements to study the characteristics of ozone diurnal variations and the upper ozone transport.

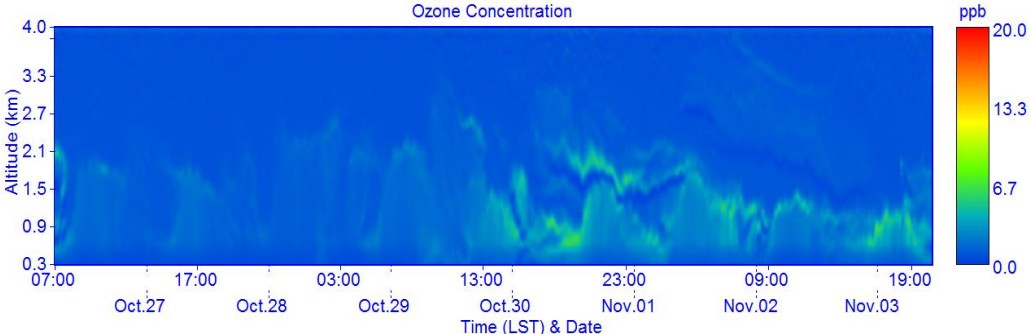

**Fig. 11.** Aerosol correction term errors when $k$ and $\mu$ changed from 1 to 0.

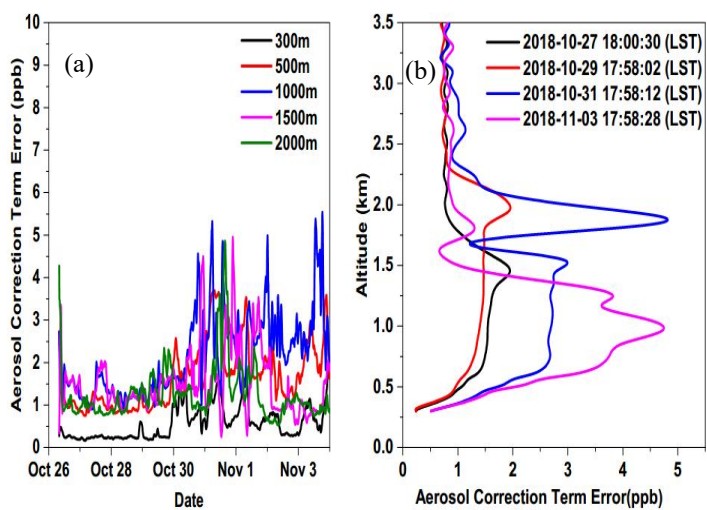

**Fig. 12.** Aerosol correction term errors when $k$ and $\mu$ changed from 1 to 1.4 at different heights and times.

## 5 Field validation with vertical observations of tethered balloon

The developed ozone lidar was deployed in a field campaign that was conducted to make vertical observations of air pollutants using a large tethered balloon platform. The campaign was carried out in December 2018 at wangdu County, Baoding City, Hebei Province, China, which is located in the center of the Beijing-Tianjin-Shijiazhuang Economic Triangle showed in Fig. 13a. It is a typical site for studying air pollution in the Beijing-Tianjin-Hebei region. The tethered balloon is equipped with a high performance mini air station (MAS-AF300 Sapiens, Hong Kong) (L. Sun et al.,2016) which can measure up to six gaseous pollutants simultaneously including $O_3$ concentration at different heights when the tethered balloon was launched. It shows reliable performance under the wide range of environmental conditions, which warrants its

application for the vertical measurement of ozone concentration under fast changing meteorological conditions. Figure 13b

shows the instrument.

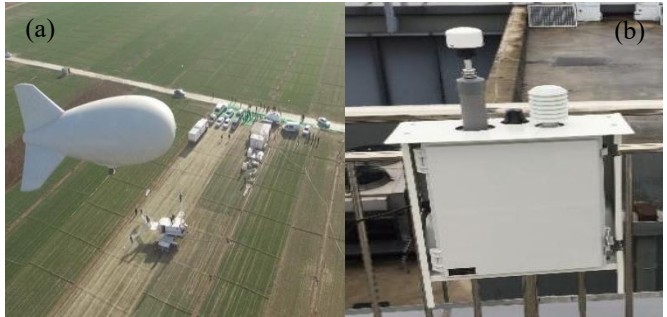

**Fig. 13 (a) Field campaign at Wangdu County during December 2018; (b) MAS-AF300 air quality monitoring system.**

During the campaign, an $O_3$ analyzer (Model: 49i, Thermo Fisher Scientific Inc., USA) was used for ground level observation. Figure 14 showed the measurement result comparisons of MAS-AF300 and Model 49i $O_3$ analyzer at 5 min

resolution during December 15 to December 16 before the field experiments. MAS-AF300 showed strong correlation to Model 49i (R2 > 0.9). The average concentration differences found were 2.3 ppb based on error analysis results. The comparison results indicated that the sensor could be accepted as a reference data source to evaluate the $O_3$ lidar performance.

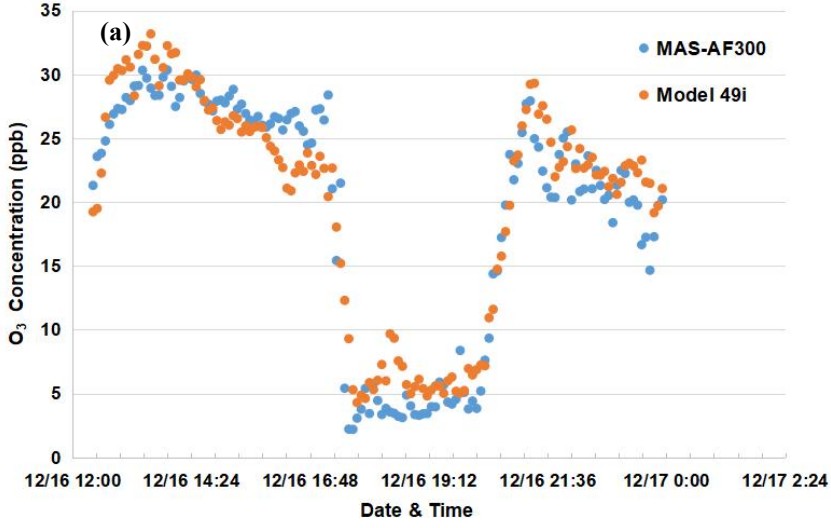

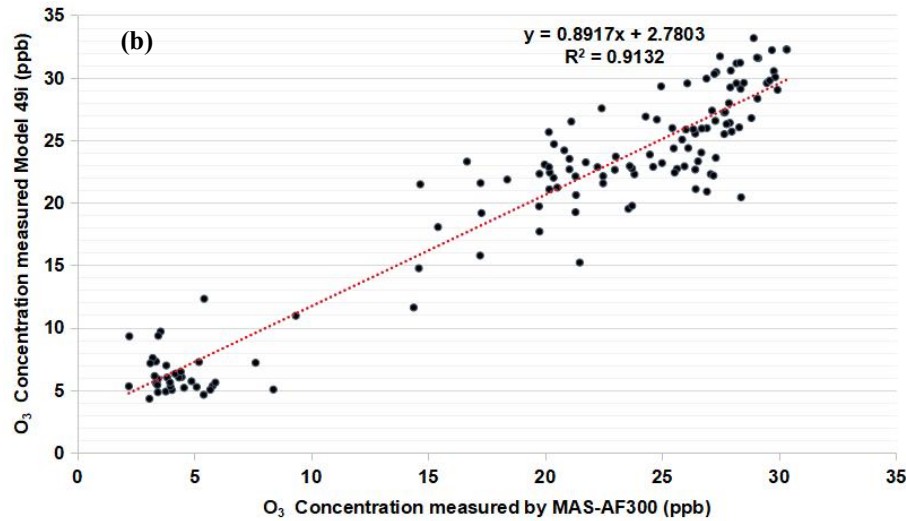

Fig. 14. Comparison of the O₃ concentration from MAS-AF300 and Model 49i (a) time series plot (b)correlation between MAS-AF300 and Model 49i.

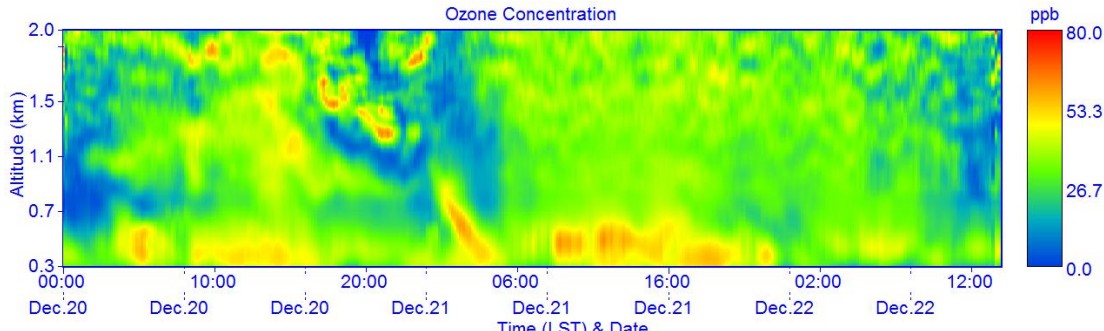

Fig. 15. Time series plot of O₃ concentrations at different heights measured by O3 lidar.

Figure 15 presents the observations of the O₃ lidar from December 19 to December 22. The vertical resolution is 100 m and the temporal resolution is 700 s. Ozone concentrations was in the ranges of 0.3 km to 2 km with an average at 34.8 ppb. The ozone concentrations observed below 700 m exhibits a significant diurnal variation pattern with high values occurring in the afternoon period. The ozone peak value at 300 m on 20th and 21st is 49 ppb and 54 ppb, respectively. However, the ozone concentration rose to 46 ppb at 3:46 am on the 21st, corresponding to 23 ppb at 19:21 on 20th. As shown in Fig. 15, ozone concentrations in the height range of 700 m~ 1000 m are rapidly mixed down to a height of 300 m, resulting in a sudden increase in ozone concentration at night.

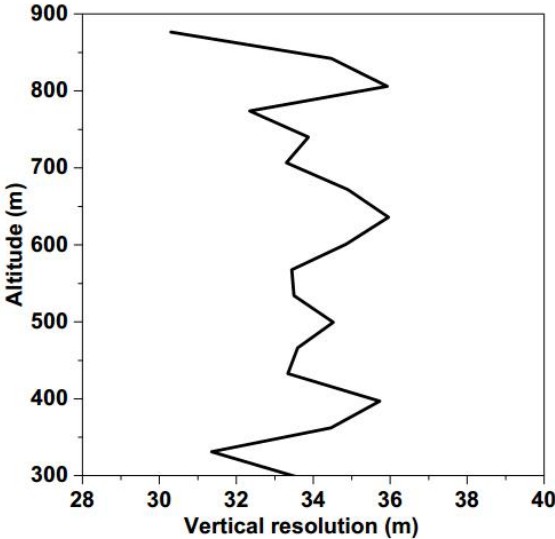

**Fig. 16. The vertical resolution of ozone data measured by captive balloon.**

The vertical profiles of ozone were obtained when the balloon was controlled to ascend or descend, and the pollutant at a fixed-height could be studied when the balloon was hovering. The maximum operating height of the tethered balloon is 900

340   m, while the lowest detection height of the ozone lidar is about 300 m, so the profiles measured by the two instruments ranging from 300 m to 900 m is mainly compared. While the ozone concentration profiles measured by ozone lidar are the cumulative averages of 400 s worth of data. The temporal resolution of the ozone data measured by MAS-AF300 is 1 min. The balloon recorded ozone data during a landing that took 25 min to 30 min. The vertical resolution of the ozone data recorded bythe balloon varied with the rise rate of the tethered balloon, as shown in Fig. 16. The average of the vertical

resolution is 33.7 m. Figure 17 shows comparisons of the $O_3$ concentration from ozone lidar measurements and tethered balloon for vertical profiles determined at different times periods on December 20 and for the time segments of ozone concentration at fixed heights of 400 m and 500 m. In general, the lidar results are very consistent with the tethered balloon observations. The relative difference is 5 ppb within in most altitude ranges and in most times at fixed heights of 400 m and 500 m. So possible reasons for the difference may be caused by the different vertical resolution and temporal resolution

between lidar and tethered balloon. In particular, as shown in Fig. 15, the ozone air mass within 800 m ~ 1000 m was transported almost to the near ground, which was confirmed by the tethered balloon at 500 m, as shown in the Fig. 17d. As can ben seen in this figure, under the influence of the descending ozone air mass, both the ozone concentration observed by the ozone lidar and the tethered balloon increased from 35 ppb at 0:00 am to approximately 50 ppb at 3:00 am at 500 m, and which then gradually fell.

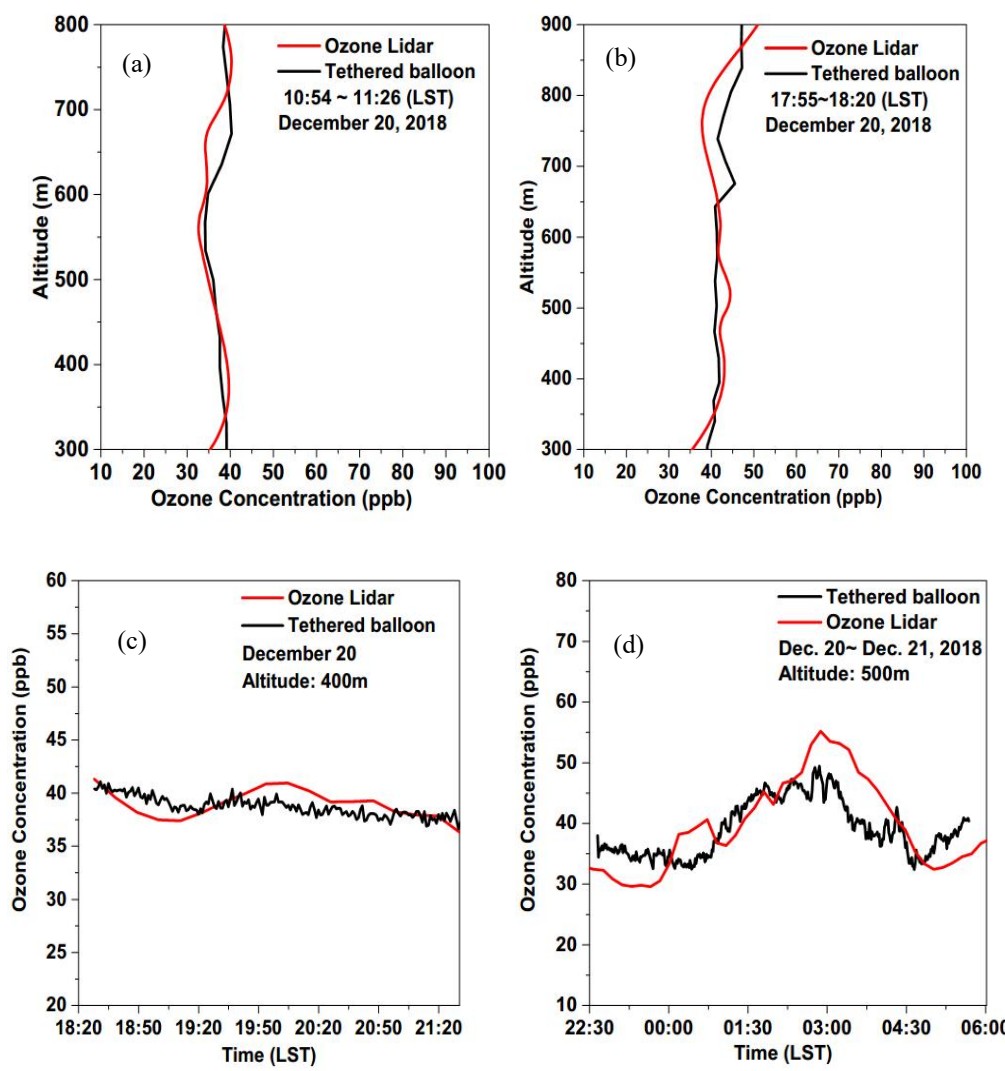

**Fig. 17. Comparison of the O3 concentrations from the tethered balloon and O3 lidar measurements for vertical profiles determined at different times on December 20 and at fixed heights of 400 m and 500 m.**

## 6 Conclusion

In this paper, a differential absorption lidar for the simultaneous observations of lower tropospheric ozone is described in detail, which was based on a single $CO_2$ Raman cell and the high-resolution grating spectrometer. A flashmap-pumped Nd: YAG laser, which provides 90 mJ output at wavelength of 266 nm and 10 Hz pulse repetition rate, is used as the pump source for $CO_2$ Raman cell. The $CO_2$ Raman cell filled with 16 bar pure $CO_2$ with 99.999% purity. The laser energy output of 276 nm, 287 nm and 299 nm are 8.4 mJ, 7.7 mJ and 4.2 mJ respectively. A 250 mm telescope and the grating spectrometer compose the lidar receiver. For signal acquisition, in order to reduce the influence of the A/D electronic noise

on the atmospheric echo signal, amplifiers of 2 times and 48 times are adopted for the same echo signal, respectively for the near altitude signal and the long altitude signal. Within the range of 500 m~1 km, the echo signals of 2 times magnification and 48 times magnification are fused using the least square method.

Take *SF* and SNR into account, below 600 m, the signals at the wavelengths of 276 nm and 287 nm were used to retrieve ozone concentration profile; Above 600 m, the signals at 287 nm and 299 nm were used. The statistical error from 200 m to 600 m gradually increased from 2.35% to 6.9%. The statistical error of ozone from 600 m to 2.7 km was basically within 3%. From 2.7 km to 3.9 km, the statistical error gradually increased to about 18%. We also evaluated the errors caused by wavelength index uncertainty. Some examples at different aerosol distributions and concentrations at Hefei are provided to illustrate the errors caused by angstrom wavelength index uncertainty while ranged from 0.6 to 1.4, the results revealed that the maximum error of the aerosol correction term was 5ppb; the error displayed a single-peak distribution.

The developed ozone lidar was deployed in a field experiment conducted with vertical profile observations using a tethered balloon. The observed lidar ozone results exhibited good agreement with those observed by the tethered balloon, confirming that the ozone lidar measurements are accurate.The bind zone of the ozone lidar is about 300m. In future work, we plan to design a 100 mm telescope to extend the observation range starting from the near surface (about 100 m) and study the exchange between near-surface and troposphere ozone.

*Data availability.* Lidar measurements are available upon request.

*Author contributions.* GF developed the methodology, designed the ozone lidar, developed the analysis code, and wrote the manuscript. YF developed the A/D data acquisition system. JH contributed to data analysis. YX performed lidar measurements. TZ and WL participated in methodology development and supervised the project. ZN supported the ozone data of MAS-AF300.

*Competing interests***.** The authors declare that they have no conflict of interest.

*Acknowledgments.* This research has been supported by the National Key R&D Program of China (2022YFC3700400), Hefei Comprehensive National Science Center and the National Natural Science Foundation of China (42005106, 41941011).

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
