# Peer review of "Tropospheric Ozone sensing with a differential absorption lidar based on single CO2 Raman cell"

_EGUsphere, 2024_

## Referee Comment (RC1)

**Review of "Tropospheric ozone sensing with a differential absorption lidar based on single CO$_2$ Raman cell" by Fan et al.**

This manuscript "Tropospheric ozone sensing with a differential absorption lidar based on single CO2 Raman cell" describes a useful differential absorption lidar and its application in ozone measurement. In general, this work is valuable and interesting, but it needs to be improved further. I recommend the paper for publication after the following comments have been addressed.

**General comments:**

1. "Considering the influences of aerosol interference and statistical error, a wavelength pair of 276 nm-287 nm is used for the altitude below 600 m and a wavelength pair of 287 nm-299 nm is used for the altitude above 600 m to invert ozone concentration." Why do you choose 600 meters as the threshold value to analyze the ozone vertical characteristic? Please explain in detail.

2. L330: It is suggested first to describe the shortcomings or problems still unresolved in this type of LiDAR, then expand to future work.

3. References are few and old. Please add more recent related references.

**Specific Comments:**

1. Line 27 "B. Koo et al,.2012" should be "B. Koo et al.,2012".Please carefully check the format of cited references throughout the manuscript.

2. Line 80: may not be correct. 'i' is not mentioned in the context. Thus 'i' should be changed to "on" or "off".

3. Line 95: 'It is' should be 'it is'.

4. Line 108 "2" in "(e-20 cm2)" should be subscript. Please check the formats throughout the manuscript.

5. Line 125: "According to (5)" should be "formula (5)".

6. Line 132: The font in Figure 2 is indistinct. It is suggested to redraw the picture.

7. Line 174: Figure 3 is small. It is suggested to adjust the size.

8. Line 215: 'In addition, The aerosol' should be 'In addition, the aerosol'.

9. Line 222: "relatively high concentration levels..." Please specify the concentration values or ranges.

10. Line 230: There is a space between them and 'aerosol' and should be deleted.

11. Line 285: Figure 12 is a bit small and distorted and it is proposed to be redrawn.

---

## Author Comment (AC1)

**Authors' Response to Reviewer #1's Comments**

Dear Reviewer,

  Thank you for your comments concerning our manuscript entitled "Tropospheric ozone sensing with a differential absorption lidar based on single $CO_2$ Raman cell" (MS NO: egusphere-2024-1853). Those comments are all valuable and very helpful for revising and improving our paper. We have studied the comments carefully and have made corrections which we hope will get your approval. Our point-to-point responses to the reviewer's comments are as flowing:

**General comments:**

1. "Considering the influences of aerosol interference and statistical error, a wavelength pair of 276 nm-287 nm is used for the altitude below 600 m and a wavelength pair of 287 nm-299 nm is used for the altitude above 600 m to invert ozone concentration." Why do you choose 600 meters as the threshold value to analyze the ozone vertical characteristic? Please explain in detail.

Reply: Thanks for the comments. Table 1ists the SF of the differential absorption wavelength pairs. Theoretically, the smaller the SF is, the smaller the influence of aerosol interference on ozone retrieval results. The SF of the differential absorption wavelength pair of 276.2 nm and 287.2 nm is nearly half of that of the 287.2 nm and 299.1 nm pair, indicating that $E_a$, $B$ of the wavelength pair of 276.2 nm and 287.2 nm is nearly half of that of the 287.2 nm and 299.1 nm. The detection of ozone at wavelengths of 276.2 nm and 287.2 nm is limited by the detection range of 276.2 nm. As shown in the figure, the signal-to-noise ratio of the 276nm signal is greater than 100 below 600m, which meets the detection requirements with sufficient signal-to-noise ratio. Above 800m, it quickly drops below 100. To accommodate different aerosol types and weather influences, and considering that aerosols are mainly distributed below a height of 600m, a height of 600m was adopted as the stitching height for the differential wavelength pair.

[Figure]

So it is only used in ozone retrieval under 600 m altitude. Above 600m, we adopted the wavelength pairs of 287.2 nm and 299.1 nm for ozone detection.

2. L330: It is suggested first to describe the shortcomings or problems still unresolved in this type LiDAR, and then expand to future work.

Reply: Thanks for the comments. "The bind zone of the ozone lidar is about 300 meters" has been supplemented.

3. References are few and old. Please add more recent related references.

Reply: Thanks for the comments. We have supplemented some references in the past six years, such as X. Chi et., 2018; X. Wang et al., 2021; M. Wang et al., 2023; Y. Qian et al., 2021.

**Specific comments:**

1. Line 27 "B. Koo et al,.2012" should be "B. Koo et al.,2012". Please carefully check the format of cited references throughout the manuscript.

Reply: Thanks for the comments. line 27, "B. Koo et al,. " has been amended as "B. Koo et.,". We have checked the format of cited references throughout the manuscript carefully.

2. Line 80: may not be correct. 'i' is not mentioned in the context. Thus 'i' should be changed to "on" or "off".

Reply: Thanks for the comments. We have supplemented "i is on or off" on line 80

3.  Line 95: "It is" should be "it is".

Reply: Thanks for the comments. "It is" has been amended as "it is". on line 102.

4. Line 108 "2" in "(e-20 cm2)" should be subscript. Please check the formats throughout the manuscript.

Reply: Thanks for the comments.

"(e-20 cm2)" has been amended as "(e$^{-20}$ cm$^2$)" on line 108.

"(e-16 cm2)" has been amended as "(e$^{-16}$ cm$^2$)" on line 108.

5.  Line 125: "According to (5)" should be "formula (5)"

Reply: Thanks for the comments. "According to (5)" has been amended as "According to formula (5)".

6.  Line 132: The font in Figure 2 is indistinct. It is suggested to redraw the picture.

Reply: Thanks for the comments. We have redrawn Figure 2.

7.  Line 174: Figure 3 is small. It is suggested to adjust the size.

Reply: Thanks for the comments. We have redrawn Figure 3.

8.  Line 215: " In addition, The aerosol" should be "In addition, the aerosol".

Reply: Thanks for the comments. "In addition, The aerosol" has been amended as "In addition, the aerosol"

9.  Line 222: "relatively high concentration levels..." Please specify the concentration values or ranges.

Reply: Thanks for the comments." relatively higher concentration levels   in the boundary layer. " has been " relatively higher concentration levels about from 0.3 km-1 to 1 km$^{-1}$ in the boundary layer. "

10.  Line 230: There is a space between them and 'aerosol' and should be deleted.

Reply: Thanks for the comments.

11.  Line 285: Figure 12 is a bit small and distorted and it is proposed to be redrawn.

Reply: Thanks for the comments. We have redrawn Figure 12.

Reply: Thanks for the comments. We have redrawn Figure 12.

---

## Author Comment (AC2)

**Authors' Response to Reviewer #2's Comments**

Dear Reviewer,

   Thank you for your comments concerning our manuscript entitled "Tropospheric ozone sensing with a differential absorption lidar based on single $CO_2$ Raman cell" (MS NO: egusphere-2024-1853). Those comments are all valuable and very helpful for revising and improving our paper. We have studied the comments carefully and have made corrections which we hope will get your approval. Our point-to-point responses to the reviewer's comments are as flowing:

**General comments:**

The manuscript presents a novel setup of ozone lidar using $CO_2$ Raman tube and its application in several field measurements. The instrument seems to be very promising. The result of the comparison between lidar and vertical measurement was fantastic. Though there are plenty of sloppy writing problems and a few technique questions, current manuscript is worthy to publish. I would recommend a major revision after addressing the following comments.

Reply: Thanks for the comments.

**Specific comments:**

1.  Line 50: 'Fuel laser'? Or dye laser?

Reply: Thanks for the comments. "fuel laser" have been amended as "dye laser" on line 48, 50, 52, 53.

2.  Line 57: $H_2$ and $D_2$

Reply: Thanks for the comments. "$H_2$, $D_2$" has been amended as "$H_2$ and $D_2$".

3.  Line 58: Lidar can not be deployed by lidar.

Reply: Thanks for the comments. "The atmospheric ozone profiling Lidar developed by National Oceanic and Atmospheric Administration (NOAA) deploys a scanning four-wavelength ultraviolet differential absorption lidar" has been amended as "National Oceanic and Atmospheric Administration (NOAA) deployed a scanning four-wavelength ultraviolet differential absorption lidar".

4.  Line 61-65: Please rephrase this sentence.

Reply: Thanks for the comments. "This lidar measures tropospheric ozone and aerosols based on Raman shift wavelengths produced by $D_2$, $H_2$ Raman gases pumped by the frequency-quadrupled Nd:YAG lasers. There are two main problems in using the $D_2$, $H_2$ dual Raman cells: One is that the $D_2$, $H_2$ Raman cells share the frequency-quadrupled Nd:YAG laser and places higher requirements on the pump laser; the other is that the overlaps of the receiver field of view and laser beam divergences are different, which enlarge the blind area of the ozone detection, whereas the ozone lidar using the $CO_2$ single Raman cell can overcome these drawbacks." has been amended as "The lidar system measures tropospheric ozone and aerosols by utilizing the Raman shift wavelengths generated from $D_2$ and $H_2$ gases. However, there are two primary challenges associated with employing the $D_2$ and $H_2$ dual Raman cells: (1) Shared Laser Resource: The $D_2$ and $H_2$ Raman cells are both pumped by the same frequency-quadrupled Nd:YAG laser. This shared resources places increased demands on the pump laser's performance and stability. The lase must provide sufficient energy to effectively pump both Raman cells. (2) Receiver Field of View and Laser Divergence Overlap: The second challenge arises from the varying overlaps between the receiver's field of view and the divergences of the laser beams for the $D_2$ and $H_2$ Raman cells. These differences can result in a larger blind area during ozone detection. The blind area refers to the region where the lidar system is unable to accurately measure ozone concentrations due to the geometric constraints of the laser beam and the receiver's field of view. This can lead to incomplete or inaccurate data regarding the ozone levels in the troposphere. In contrast, an ozone lidar system utilizing a $CO_2$ single Raman cell has the potential to address these issues. The single Raman cell design simplifies the system by eliminating the need to manage two separate Raman cells, thereby reducing complexity and the need for Nd:YAG. Furthermore, the single Raman cell system may offer a more consistent overlap between the receiver's field of view and the laser beam, which can help to minimize the blind area and enhance the accuracy of ozone detection.".

5. Line 108: Nd:YAG. The sentence also needs to be improved.

Reply: Thanks for the comments. "Nd: YAGA quad-frequency laser pumped single D2 Raman tube generates first-order stokes light and second-order stokes light, corresponding to the differential absorption wavelengths of 289 nm and 316 nm." has been amended as "The Nd:YAG quad-frequency laser, when used to pump a single $D_2$ Raman tube, generates both first-order and second-order Stokes light. These correspond to the differential absorption wavelengths of 289 nm and 316 nm, respectively.".

6. Line 118: Give the reason why 276nm laser can only be used under 600m.

Reply: Thanks for the comments. Due to the strong absorption of ozone at 276nm and the strong atmospheric backscatter at this wavelength, the detection height of the 276nm wavelength signal is limited. As shown in the figure, the signal-to-background ratio of the 276nm signal is greater than 100 below 600m, which meets the detection requirements with sufficient signal-to-noise ratio. Above 800m, it quickly drops

below 100. To accommodate different aerosol types and weather influences, and considering that aerosols are mainly distributed below a height of 600m, a height of 600m was adopted as the stitching height for the differential wavelength pair.

[Figure]

7. Line 127-129: $SO_2$ interference should be addressed in detail, here and in section 4

Reply: Thanks for the comments. The table below analyzes the extent of interference from $NO_2$ and $SO_2$ gases. The interference from $NO_2$ at the 276.2nm/287.2nm wavelength pair and the 287.2nm/299.1nm wavelength pair is 0.98% and 3.5% of the $NO_2$ concentration, respectively, which can be neglected. The impact of $SO_2$ on ozone is more significant, with impacts of 8.9% and 34.7% of the $SO_2$ concentration at the two wavelength pairs. The typical environmental concentration of $SO_2$ is a few (ug/m³). If assessed at 10 ug/m³, its impact would be approximately 0.89 ug/m³ and 3.5ug/m³, which is relatively small compared to other sources of error and is therefore usually not considered. Another factor is that if the interference effect of $SO_2$ concentration is taken into account, it would also be necessary to measure the spatiotemporal distribution of $SO_2$ in real time, which would undoubtedly be very costly.

| Wavelength pairs (nm) | $SO_2(e^{-20}cm^2)$ | $NO_2(e^{-20}cm^2)$ | $O_3(e^{-20}\ cm^2)$ | $NO_2$ interference | SO2 interference |
|---|---|---|---|---|---|
| 276.2/287.2 | 30 | 3.3 | 335.43 | 0.98% * $NO_2$ Concentration | 8.9% * $SO_2$ concentration |

| 287.2/299.1 | 52.8 | 5.4 | 152.21 | 3.5%*NO₂ concentration | 34.7% * SO₂ concentration |
|---|---|---|---|---|---|

8. Line 147: How about the rest power of the pumping laser? Does 266nm laser still exist in the output laser beam?

Reply: Thanks for the comments. The 266nm laser still exist in the output laser beam, with the residual single-pulse energy at 266nm being approximately 14mJ.

9. Line 166-167: Please rephrase this sentence. The reason to use JGS1 quartz should be mentioned.

Reply: Thanks for the comments. "JGS1 quartz material is adopted in the three sets of high-reflection flat concave mirrors." has been amended as "The three sets of high-reflection flat concave mirrors are constructed using JGS1 quartz material, which is chosen for its superior optical properties and resistance to laser damage, ensuring high reflectivity and durability in the system."

10. Line 199: What is the meaning of 4000 laser pulse? Total measurement time?

Reply: Thanks for the comments. 4000 laser pulses correspond to the raw data acquisition time. Given that the laser operates at a frequency of 10 Hz, 4000 laser pulses equate to a data acquisition period of 400 seconds.

11. Line 204: How to determine the statistical error should use more serious statistics. Currently the error estimation method as well as clear conclusion of the estimation are missing. The same for section 4.2.

Reply: Thanks for the comments. The statistical error is

$$
\frac{\delta(N(z))}{N(z)}
$$
$$
= \frac{1}{2\Delta\delta\Delta z N(z)}\sqrt{\frac{\delta P(\lambda_{on}, z)^2}{P(\lambda_{on}, z)^2} + \frac{\delta P(\lambda_{on}, z + \Delta z)^2}{P(\lambda_{on}, z + \Delta z)^2} + \frac{\delta P(\lambda_{off}, z)^2}{P(\lambda_{off}, z)^2} + \frac{\delta P(\lambda_{on}, z + \Delta z)^2}{P(\lambda_{on}, z + \Delta z)^2}}
$$

The statistical error of ozone lidar data is inversely proportional to the absorption cross-section difference, the difference distance, the unknown gas concentration, and the SNR of the ozone data. The statistical error of the ozone lidar is related not only to the hardware of the device but can also be considered constant in the short term, aside from its dependence on atmospheric conditions and solar irradiance. Generally, due to the influence of solar irradiance, the signal-to-noise ratio (SNR) of daytime signals is typically lower than that of nighttime signals. During the observation period from October 26 to November 3, 2018, the SNR of the 299.1nm signal remained essentially stable as shown in the blow figure. Therefore, the statistical error of ozone at 11:00 to 12:00 on October 26, 2018, was used to analyze the performance of the ozone lidar. It

is important to note that an ozone lidar is an in situ measurement device that is closely related to atmospheric conditions, and its SNR can drop sharply during extreme weather conditions such as rain or fog, leading to a significant increase in statistical error.

[Figure]

12. Line 229: I can not find where the aerosol extinction coefficient is less than 0.3 km$^{-1}$ in Fig.7.

Reply: Thanks for the comments. "0.3 km" has been amended as " 0.5km".

13. Line 236: 'times'?

Reply: Thanks for the comments. " Figure 8(a) shows the times of the aerosol correction term" has been amended as "Figure 8(a) shows the aerosol correction term"

14. Line 251: Something missing before 'therefore'.

Reply: Thanks for the comments."It indicated that the aerosol correction term changes rapidly in the process of aerosol transport and when the boundary layer aerosol concentration is high" has been amended as "The analysis indicates that the aerosol correction term exhibits rapid fluctuations during the transport of aerosols, particularly when the concentration of boundary layer aerosols is elevated."

15. Line 285: Fig.12 is not readable.

Reply: Thanks for the comments. We have redrawn fig.12.